# Educating in History: Thinking Historically through Historical Reenactment

José-Manuel González-González *, Jesús-Gerardo Franco-Calvo and Darío Español-Solana

Argos Group, Department of Specific Didactics, Faculty of Education, Universidad de Zaragoza, Calle Pedro Cerbuna, 12, 50009 Zaragoza, Spain; jgfranco@unizar.es (J.-G.F.-C.); despanol@unizar.es (D.E.-S.)
* Correspondence: joseman@unizar.es

**Abstract:** This paper aimed to identify trends in the scientific literature that relate the link between two concepts: historical thinking and historical reenactment. The definition of both concepts and their commonalities were examined. Convinced that History instruction and Heritage education could improve new methods and techniques, and aware of the benefits of reenactments in active learning and participation in and outside the classroom, we came to the obvious conclusion that merging both aspects is a must and should be disseminated. We also analyzed the presence of second-order concepts in reenactment practices and how they are addressed by actors and spectators. Reenactments foster the acquisition of critical thinking by citizens through education; their quality, however, must be improved through research and didactics—didactics based on reenactment that help us value the past and the traces still present in local areas. Local and global identity and heritage, emotions, reproduction of objects, the use of sources, relevance, empathy, multiperspectives, causation, communication, the relationship between past and present, and the sustainable economy proposed by the 2030 Agenda, are all aspects that should take center stage in turning this phenomenon into a living and lasting history as an experience.

**Keywords:** historical thinking; historical reenactment; reenactment thinking; history teaching; sustainability; heritage education





## 1. Introduction: Historical Reenactment: Definition, Objectives, Characteristics, and Methods

The most widespread definition of historical reenactment—or living history—is the practice of reconstructing uses, customs, material culture, and aspects of the past based on strictly scientific guidelines to achieve objectives related to cultural dissemination and education (Español-Solana 2019c). The adequacy of this tool to provide a historical approach—and, therefore, historiographical research—as a result of experimentation (Español-Solana 2021; d'Oro 2004; De Certau 1984; Stueber 2002) should also be added to the above description. Stueber's theses are based on the philosophical perspective of historical thinking, while D'Oro confronts the theoretical views on historical reenactment held by Davidson, Gadamer, and Collingwood (Retz 2018). Consequently, reconstructing the past to understand it is not only confined to its material culture (Samida 2019), but also to practices, aptitudes, and linguistic uses, among other areas, always with scientific and academic support (Español-Solana 2019b). With this practice, we aim to achieve several objectives:

- show the importance of historical reenactment as an educational tool.
- identify the relation between historical thinking and historical reenactment.
- analyze the presence of second-order concepts in reenactment practices.

Although history can neither be experienced, nor directly observed or replicated, there are alternatives to understand it and feel it as much as possible through live or computer

simulations that partially overcome the initial abstraction (Cardona Gómez and Feliu Torruella 2014; Kneebone and Woods 2014).

Historical reenactment, as a kind of reconstruction of History, aimed at researching or educating, implements a process linked to the historical method, and, in most cases, it adds another element as well: knowledge transfer or dissemination. This reenactment could be surrounded by a praxis linked to experimental archaeology (Meylan 2013), to which it is inherently related (Del Barco Díaz 2010). However, historical reenactment is characterized by a univocal inclination toward dissemination and didactics (Cózar Llistó 2013), which is captured in the documentation and reconstruction process of the material culture.

Arcega Morales (2018) underscores eagerness to learn about history, liking for performing arts, and cost issues as reasons that might explain why this phenomenon is becoming increasingly popular. Tourism, financial concerns, the need to know, the search for identity roots in the community (López Cruz and Cuenca López 2014; Corral Lafuente 2019), association involvement, or aims relating to enhancing heritage (Rojas Rabaneda 2019) are some of the goals that may arise when planning activities of this type; however, they should not be considered historical reenactments if the objectives of socializing knowledge or being a research tool are not met (Español-Solana 2019b).

In this respect, the concept of historical reenactment has deteriorated as two completely different models are termed the same way (Español-Solana 2019b; Aquillué Domínguez 2019; Rojas Rabaneda 2019; Español-Solana et al. 2020). On the one hand, there are historical festivals and, on the other, historical reenactments, in the strict sense. Although the difference between both models is known in most of Europe, in Spain, in line with a trend in southern Europe, the distinction has narrowed to the point where the former has adopted the name of the latter. As a result, activities that are actually historical festivals are labeled historical reenactments, possibly in an attempt to make the contents sound more rigorous (Balbás 2019; Rojas Rabaneda 2019). This conceptual inaccuracy involves debatable stances in the opinions seen in part of the literature on the subject. For instance, Jay Anderson analyzed living history museums from the 1890s onwards and identified three types of reenactors, based on their capacity for more or less immersion (Anderson 1984).

As has been mentioned, although historicist evocations such as festivals, markets, processions, or plays may contain heritage and historical elements, these are extremely stereotyped as they are not aimed at education or research and, therefore, they lack scientific methodologies ensuring they are considered as quality scientific or educational praxis. The reason is that they respond to commercial, tourism, or other types of objectives, and do not approach historical facts based on experience or historiography, nor do they construct accurate educational discourses. The disconnect between academia and museology in these historicist manifestations must be understood as a consequence of this disparity of objectives and, therefore, of results.

Nevertheless, the experiential factor can, and must, be subjected to an epistemological examination that, in the worst scenario, would invalidate its effectiveness for research and for thinking historically (Agnew 2004, 2007; Lévesque 2009; Van der Plaetsen 2014; Retz 2018). These folkloric manifestations may also be studied on the basis of the socializing or introductory function of historical knowledge, as well as local and regional identity.

This problem has led organizations to review their programming to try to differentiate themselves from others. In recent years, historical festivals and reenactments have changed from being viewed by specialists as representations of the past aimed at entertaining tourists to becoming models seeking to socialize knowledge on history and heritage sites (Sebares Valle 2017; Español-Solana et al. 2020). This is where we can find the second model that several authors propose (Brædder 2019; Español-Solana 2019a) related to academia, research.

Although collaborations between heritage assets and historical reenactments take place occasionally, all researchers agree that such a union should be essential, as it is one of the ways of enhancing heritage (Egberts 2014; Franco-Calvo et al. 2020; Felices de la Fuente and Hernández Salmerón 2019; Solé 2019). In fact, recent studies advocate the indissolubility

between the latter and the reconstruction of the past, backing their arguments with audience studies (Español-Solana and Franco-Calvo 2021a). This enhancement allows the increase of cultural tourism, which optimizes heritage and natural resources, respects local identity and traditions, and generates socio-economic benefits in the community of origin.

For this article, we have followed the PRISMA model. For this state of the art we have used scientific databases and have found these results:

- *Web of Science* (https://www.webofscience.com/wos/woscc/basic-search) (accessed on 1 December 2021):

    - 724 articles for "reenactment", 115 from History and 25 from Art;
    - 22 texts for "historical reenactment";
    - 689 results for "historical thinking", 524 articles, 225 from the field of "educational research", 177 open access;
    - 5 results for "historical", "thinking" and "reenactment" keywords related to our field.

- *ERIC* (https://eric.ed.gov/) (accessed on 9 December 2021):

    - 176 results for "reenactment", 111 journal articles, 26 related to "History instruction";
    - 17 texts for "historical reenactment";
    - 435 results for "historical thinking", 325 journal articles, 150 from the last ten years;
    - 3 results for "historical", "thinking" and "reenactment" keywords.

- *Dialnet* (https://dialnet.unirioja.es/) (accessed on 21 November 2021):

    - 63 documents for "reenactment", 29 from Humanities, 11 from Art studies, and 10 from Social Sciences;
    - 115 for "recreación histórica", 56 from Humanities, 33 from Social Sciences, and 28 from Art. 14 documents for "historical reenactment".
    - 433 results for "pensamiento histórico", 282 journal articles, 121 from Education, and with full texts. 204 results for "historical thinking", 20 with full texts in English, 9 in Portuguese.
    - 56 results for "pensamiento" "histórico" and "recreación", 0 for "historical", "thinking" and "reenactment" keywords.

We finally have collected those more relevant to write this article.

## 2. Discussion

### 2.1. Developing Historical Thinking through Historical Reenactment

Thinking historically is one way to achieve quality education in history instruction, especially in secondary school. As a key competence in our subject, a priority objective is for pupils to develop critical, contextualized, source- and perspective-based, empathetic, and informed thinking. In essence, it means pupils approach history with a historian's mentality and techniques, not an easy task, even for adults; however, pupils are not expected to become historians, an aspect typical of the university stage. This leap in quality, if this type of reflective thinking is attained, results in in-depth and reasoned learning, in other words, founded on sound judgment.

Thinking historically helps young people attain comprehension, quality knowledge, and competencies (Domínguez Castillo 2015; Gómez and Chapman 2016). Knowing which events were relevant, what has changed and what has not, how the past is reflected in our surroundings, and how certain events influenced the present is crucial for this process. Interpretation and analytical skills are essential in psychosocial maturity.

There are many authors who have managed the study of historical thinking, such as Shemilt (1978; History 13–16, 1980); Lee (History Teaching and Philosophy of History, 1983; 2000; 2005); Seixas (Historical Understanding among Adolescents in a Multicultural Setting, 1993; 2006); Vilar (Pensar históricamente, reflexiones y recuerdos, 1997); Wineburg (Historical Thinking, 2001; Why Learn History, 2018); Barton (Historical Thinking in



the Elementary Years: A Review of Current Research, 2004; 2017); Lévesque (Thinking Historically, 2009); or Seixas and Morton (The Big Six, 2013).

As these authors explained, historical thinking comprises several second-order aspects or concepts, or metaconcepts; cognitive skills typical of our discipline rather than repetitive skills. The historical consciousness every individual has and develops is another step to be implemented in teaching-learning processes. Although it was the British, Americans, and Canadians who researched these subjects more in-depth, professionals of other nationalities (German, Dutch, and Spanish) have also conducted studies in this area (Domínguez Castillo 2015; Rodríguez-Medina et al. 2020).

"Yet to understand the history discipline as a whole, Lee (1983) stressed the importance of drawing on procedural history, or second-order historical concepts, in conjunction with substantive history when undertaking historical inquiry" (Shaw 2021, p. 85). Wineburg questioned whether knowledge should form the basis of learning, as it does in the renowned taxonomy pyramid by Bloom, who, although much followed in didactics, is not at all recommendable for the subject of history, since critical thinking may turn this foundation upside down: "Putting knowledge at the base implies that the world of ideas is fully known and that critical thinking means gathering accepted facts in order to render judgement" (Wineburg 2018, p. 92).

Nevertheless, the study of English and Spanish secondary education curricula has determined that, while the 2008 English legislation provided that almost 50% of the content must be based on second-order concepts, in the 2015 Spanish legislation it was only 5%, although in previous Spanish laws historical thinking had more presence (Gómez and Chapman 2016, pp. 437–39). A recent study including the Portuguese curriculum has also underscored that legislative references to historical thinking were far more evident in the English case than in the other two countries analyzed (Santisteban et al. 2021, p. 21). Similarly, in Canada, Haskings-Winner has stated the difficulty of applying historical thinking in classrooms: "Sometimes the process of doing history and thinking historically can get forgotten, or teachers return to what they think history education is: lectures or the dissemination of facts, and, more recently, the use of technologies such as PowerPoint and smartboards to show pictures that reinforce talking points" (Sandwell and von Heyking 2014, p. 284). In Latin America, some countries include some of these meta concepts in their curriculum (Domínguez Castillo 2015, pp. 60–62).

In Spain, several books (Domínguez Castillo 2015), articles, and doctoral theses on history didactics emphasize the interest in competence development of historical thinking; this topic has aroused the most interest among Ph.D. students, as shown by a thorough list recently analyzed (Chaparro Sainz et al. 2020, p. 95). High-impact publications have demonstrated the importance of combining consciousness and historical thinking with the ethical dimension (Cavanna et al. 2021), and countless articles have exhibited an interest in addressing these aspects (Retz 2018; Rodríguez-Medina et al. 2020).

Historical reenactment also provides whoever experiences it with an in-depth understanding of history and facilitates informed and well-understood thinking as it presents a living history (Anderson 1984) that is also scientific, more approachable, and uses images and actions that will aid said reasoning. History has to be interpreted and imagined, and it requires narrators and reenactors. The relationship with nearby familiar elements researched or experienced with this reenactment helps consolidate second-order concepts, as shown in recent studies (Shaw 2021).

The use of historical reenactment in the classroom dates back to the 1960s in the United Kingdom (Retz 2018) and to the 1980s in the United States. Several American scientists highlighted that these activities or events had more impact on learning than dramatization, since they enabled an almost full immersion and empathic understanding of the historical moment being represented (Turner 1985). If performed with the required preparation and time, reenactment ensures pupils gain a broad, in-depth knowledge based on relevant sources and on adopting a historical perspective: "Simpler reenactments can involve students and teachers intellectually. Such intellectual involvement promotes

better attention, interest, concern, reaction, and evaluation. [...] This activity can be an essential power for learning history and understanding contemporary social situations" (Turner 1985, p. 221). Turner and other authors in their more recent works advise using appropriate themes for young people that are related to their identity and local or regional heritage; simple, clear events with well-known characters or phenomena that have an impact on their emotions and end satisfaction, and pay attention to the importance of the location, the facilities, the program, the atmosphere or the ambience (López Cruz and Cuenca López 2014; Carneiro et al. 2019).

Nowadays, digitalization and new technologies, such as green screens, have enabled the introduction of reenactments in the classroom (Sheffield and Swan 2012), which is highly recommended for school settings. The process becomes far more GOALeffective if we add auditory immersion to visual immersion, since historically recreating sound may serve as an educational aid, as a primary source and emotional stimulus (Carneiro et al. 2019), by listening to or playing music (Goering and Burenheide 2010).

A recently published monograph on the relationship between violent pasts and reenactments analyzes their popularity and how this relates to the capacities and skills to think or make think historically that can be found in reenactors and spectators, and on the possible link between formal and informal education that come together as a result of these initiatives (Grever and Nieuwenhuyse 2020).

> "In the 1970s, the term 'public history' gradually became acknowledged as a historical sub-discipline. Public history became institutionalized with the founding of graduate programmes, specific journals, and associations. [...]"

> "As the observer of the past is always situated in a present which influences his/her view of the past, historical thinking is also about building an understanding of and reflecting on the complex relationship between past, present, and future. [ . . . ] In formal history education at secondary schools, historical thinking has increasingly been adopted as the main aim of the school subject. In the history curricula of provinces, states, and countries such as British Columbia (Canada), California (United States), Flanders (Belgium), the Netherlands, Sweden, England, and Finland, historical thinking and historical reasoning occupy the centre stage (see particularly the related chapters in Metzger and Harris 2018)". (Grever and Nieuwenhuyse 2020, pp. 487–89)

Although we need to distinguish between professional and amateur reenactment (Brædder 2019)—or, as already mentioned, between reenactment and evocation or commemoration (Español-Solana et al. 2020)—we cannot ignore that some theorists oppose the idea that reenactments may be an appropriate educational tool for the subject of history, or for true knowledge of our discipline, since they often lead to disappointment, or dilute what is represented (Cook 2004; Handler and Saxton 1988). While no simulation of the past can obviously be perfect, some movies for television and theaters have paid special attention to production and language, and have even consulted and hired expert historians to make reenactments as close to actual events as possible. Nevertheless, according to Cook, they will never be the same, since many emotional, contextual, behavioral, and attitudinal issues are impossible to reproduce: "The challenge is to find a way of illustrating critical engagement with the past in a manner that captures the imagination of a lay audience—an audience that may well be eager for dramatic narrative and impatient with ambiguity and contention" (Cook 2004, p. 495).

For this reason, some authors that have interviewed professional reenactors specialized in several historical stages have found they are inspired by a passion for learning focused on others, on themselves, and on scientific material itself.

> "Against these criticisms, scholars with a more social-constructionist approach have ventured that the essentialist distinction between authentic and inauthentic is untenable [...] Reenactment may have a genuine investigative dimension in which the quest for authenticity is not solely about dramatising an already well-

known past but generating new knowledge through the activity itself (Crang 1996, pp. 419–20; Cook 2004, pp. 487–88)". (Brædder et al. 2017, pp. 172–73)

It is authenticity (Handler and Saxton 1988; Brædder et al. 2017), or its most scientific approximation, that makes these reenactments truly good learning practices and enables them to attain considerable verisimilitude or legitimacy, auctoritas or prestige. In conclusion, it involves more in-depth knowledge than what can typically be taught in the classroom (Wineburg 2018, p. 100).

*2.2. Socializing Knowledge through Historical Reenactment and Its Didactic Use*

Researchers unanimously support the idea that history should be reconstructed with the most rigor—by avoiding anachronism or presentism—, with fixed criteria and clear concepts regarding the actions and materials to be used, and without ruling out the possibility of documenting the reconstructive process based on prior research in line with scientific methods—as occurs with experimental archaeology—to serve as the foundation for the process (Egberts 2014; Del Barco Díaz 2010; Cózar Llistó 2013; Robinson and Yerbury 2015; Español-Solana 2019b; Felices de la Fuente and Hernández Salmerón 2019; Jardón Giner and Pérez Herrero 2019; Corral Lafuente 2019; Agnew and Tomann 2019).

On the one hand, collective memory, knowledge socialization, and the participatory learning of history and heritage are some of the drivers of this discipline and are still hard to find in some events and projects. Although a didactic methodology is not applied at all times, there is a will to educate based on research results (Rojas Rabaneda 2019). On the other, we refer to a didactic component that, when inserted into the training method of new teachers, through the theatrical representation of scenes from the past, offers them the chance to work on new cognitive skills in their pupils, such as stimulating their imagination and creativity, adopting their own identity within the community, or relating and promoting social attitudes and respect for heritage (De Paz Sánchez and Ferreras Listán 2010; Sandwell and von Heyking 2014); in short, adding values related to pupils' comprehensive education. Furthermore, the didactic use of dramatization is unquestionable, by adopting a global conception that can result in a complete historical reenactment. The adoption of drama techniques in the classroom will depend on pupils' interpretive skills. It will be further outlined as it develops into a series of specific forms: symbolic play, representation of roles, and theater (Motos Teruel and Navarro Amorós 2003; Kneebone and Woods 2014). It is no less true, however, that the use, or abuse, of dramatization in non-formal didactics does not encourage interactivity between reenactors and audiences, a key element in the educational process (Español-Solana 2019b).

From this point, we should explore the main methodological principles involved in inserting historical reenactment practices into formal settings (Robinson and Yerbury 2015) using historical thinking as a reference framework at all times (Lévesque 2009). This is crucial for understanding how important a role historical reenactment can play in history and heritage didactics. As Español-Solana (2019b) points out, we have to overcome the reductionism of materiality in order to convey broader paradigms linked to explaining ideas related to the historical time, such as change and continuity or causes and consequences (Barton 2017), among other second-order concepts. In fact, practice needs discourse, explanatory, and operational resources that transgress mere reenactment of material culture, since, otherwise, the educational practice we establish will have a modest impact. Along the same lines, some authors advocate the convenience of using an approach focused on experimentation and historical thinking (Agnew 2004, 2007; Lévesque 2009).

Empathy to put ourselves in the shoes of historical figures and infer their reasons and reactions (Hernàndez Cardona 2001; Retz 2018; Solé 2019) is also indispensable, as is emotion (Carneiro et al. 2019) in both reenactors and spectators (Español-Solana 2019b). The practice of drama, however, involves developing and controlling emotions, consciously exploring feelings and moods, pursuing Goleman's emotional education (Motos Teruel and Navarro Amorós 2003) and allowing the development of intrapersonal intelligence (Gardner 1993).

Teaching for comprehension also helps attain in-depth and significant knowledge (Ishee and Goldhaber 1990). To this end, new content should be linked with subjects' concepts and experiences, and their personal and social memory; this can be achieved by planning reenactment-based methods with useful undercurrents and experiential elements (Jiménez Torregrosa and Rojo Ariza 2014) that awaken pupils' curiosity and improve and expand their cognitive skills. Nevertheless, we cannot ignore that pupils are placed center stage in learning and that they will construct their own rational structures alongside their peers (Motos Teruel and Navarro Amorós 2003). All these aspects are fundamental in order to instill knowledge in pupils using active, constructivist, and discovery learning methods. In fact, historical reenactment—following the hands-on, minds-on, and hearts-on didactic principle—establishes methodologies in which pupils solve problems through research (Cardona Gómez and Feliu Torruella 2014; Rivero Gracia and Pelegrín Campo 2019; Felices de la Fuente and Hernández Salmerón 2019). All this involves thinking processes and is linked to skills such as reasoning, divergence (Hernàndez Cardona 2001), adaptation, and flexibility. The content to be taught needs to be problematized, while avoiding linking concepts or historical data that are not very attractive for learners (Español-Solana 2019b).

Another key factor in this process is creativity (Hernàndez Cardona 2001; Felices de la Fuente and Hernández Salmerón 2019), understood as interacting with the audience rather than relying on closed scripts (Español-Solana 2019b). Although it may prove controversial, some authors defend compatibility between flexibility and rigor (Motos Teruel and Navarro Amorós 2003), since both are needed for good historical reenactment. Pupils may also experience uchronic situations (Hernàndez Cardona 2001), or disciplined historical imagination (Solé 2019), which are perfectly valid in some contexts.

Cooperative work is another strength of this methodology (Robinson and Yerbury 2015). It promotes collective feeling and social interaction. Historical reenactment normally occurs in a group, working on the skill of connecting with other participants, which favors the development of interpersonal intelligence (Gardner 1993). In addition, designing and implementing a reenactment pose a challenge that encourages critical thinking (Thelen 2003).

From the perspective of heritage didactics, including methodologies based on reenactment fosters individuals' relationships with these assets regarding belonging, ownership, and identity (Fontal 2008; López Cruz and Cuenca López 2014), which, in turn, lead to more involvement and respect (Rivero Gracia and Campo 2015).

It is very positive that historical reenactment can be carried out directly on the scene, where it takes place. Heritage sites are ideal for explaining and understanding historical events and periods (Sebares Valle 2017), and contemplation and experimentation play an extremely important role in generating emotions and feelings that help us recognize the asset and identify with it (López Cruz and Cuenca López 2014; Felices de la Fuente and Hernández Salmerón 2019). This interest brings us closer to this heritage so we can overcome that image of opacity typical of cultural assets (Bardavio and González Marcén 2003).

A clear and definitive commitment must be made, by the different administrations and historical spaces, programming historical recreations as a way of socializing knowledge, as a way to improve and respect this local historical heritage. For this, we propose to flee from that mass and structured tourism that generates negative effects at all levels. It is necessary that quality mark the success or failure of an activity of this type, but rather the transmission and democratization of knowledge, in a community eager to obtain it, but seeking economic viability that implies prosperity of the territory and quality employment. For this, quality activities must be designed, with rigor and entertainment that provide the greatest cultural satisfaction to the visitor, trying to involve the local population in the whole process. But for a historical recreation project to be considered sustainable, from a tourism and social point of view, it must also take into account other aspects such as pollution, waste produced, or social discrimination, among many other factors of development settled by the 2030 Agenda. The 17 SDGs (Sustainable Development Goals) marked the way to take action

just for education, equality, economic growth, improving communities, and responsible consumption. Governments, companies, and reenactors should lead these civic politics.

### 2.3. Some Examples of Historical Reenactment Worldwide

Besides this general approach to certain methodological factors, we must consider the advantages this cultural industry currently offers in non-formal settings, which are easy to include in any historical and heritage education program. There are now hundreds of Spanish and international historical reenactment projects combining all the necessary factors to construct an educational event, as outlined above (Figure 1). When planned in advance by teachers, they can be visited by groups of schoolchildren and form part of educational material and practice. Many of these projects are designed to offer a wide range of didactic possibilities, since they are suitable for a variety of audiences with differing educational levels. Some examples are outlined below.

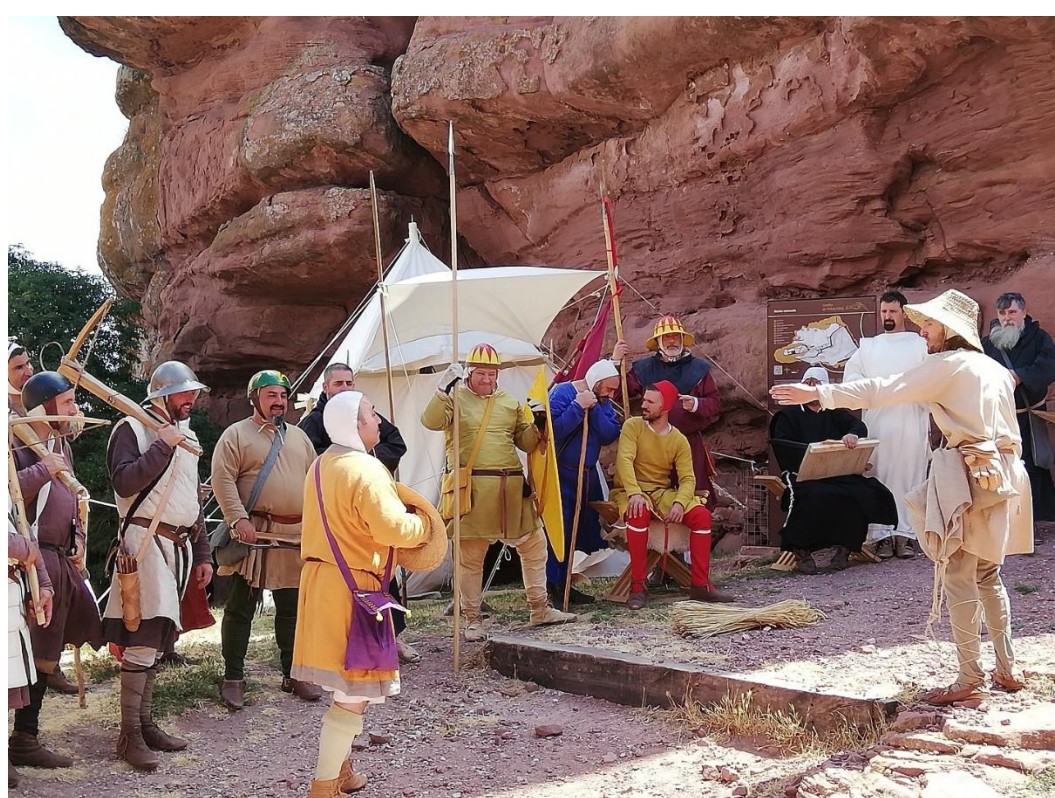

**Figure 1.** Medieval reenactment in Aragón, Spain.

American Civil War reenactments are quite common. The Battle of Olustee (Florida), for example, is reenacted yearly with specialists and historians coming from all over the country to mark this historical event. One day is completely dedicated to students so they can interview the reenactors. Another example is the Battle of Gettysburg, which has brought together up to 30,000 reenactors. In Virginia (USA) there is a living museum called Colonial Williamsburg. It is a long-term project reproducing an eighteenth-century colonial city where visitors immerse themselves in a didactic space reconstructing all aspects of civil life in this period. It is similar to the British model reproduced at the Jorvik Viking Centre. Jorvik allows visiting the site reconstructed with areas of the old Viking settlement of York. This project has revolutionized the world's panorama of how to reconstruct—down to the last detail—a phenomenon as complex as medieval military architecture, by showing the entire process. The largest and oldest association of reenactors, The Sealed Knot, is also in England. It was founded in 1968 by a group of soldiers who dress in seventeenth-century fashion.

In the Pacific, several maritime reenactments have been held with the help of the Polynesian Voyaging Society consisting in building ships and canoes to sail the ocean in a traditional manner. These reconstructions have had a considerable impact worldwide thanks to the documentaries on the topic and to the fact that, for decades, they were conceived as real research or technological and anthropological studies, despite the controversial views they elicited about their authenticity or the consideration toward native populations (Scanlan 2017).

In Russia, the Battle of Borodino, one of the most important events of Napoleon's French invasion of Russia, which happened in 1812, is reenacted yearly on the first Sunday of September. Another famous reenactment is that of the Battle of Kulikovo, that took place in 1380, which in 2016 held twenty consecutive editions.

In Europe, there are paradigmatic models such as Biskupin in Poland, which is the reconstruction of a settlement on an Iron Age site dating from the eighth century BC. This archaeological park combines activities for the general public through historical reconstruction and experimental archaeology, while it is still studied by archaeologists and historians as part of the dissemination offering. The Eketorp site in Sweden, dating from the fifth century, is similar; in this case, it is a mixed model that reconstructs ways of life and material culture. The reconstruction of the Iberian Citadel of Calafell and the transfer and subsequent reconstruction of the town El Cabo in Andorra, both in Spain, allow walking through streets and homes as they would have been in the pre-Roman period, thereby favoring historical reenactment as a means to spread awareness of the past. These examples fall under "open-air" or "open" models, which, following an increasingly popular philosophy, are pervading many management designs for heritage assets. This approach makes visitors participate in the work of archaeologists, architects, and specialists. These open-air museums include those known as archaeodromes, such as the Archeodromo di Poggibonsi in Italy, linked to the University of Siena. Major battles have also been reenacted in Europe, such as Waterloo (Belgium) and Bailén (Spain), part of the Napoleonic Wars, or the Battle of Hastings (United Kingdom) or the Normandy landings (France), from other periods.

The European Union has financed didactic actions as part of the Socrates program, fostering communication between school and society (De Paz Sánchez and Ferreras Listán 2010, pp. 531–32). In Portugal, the work on Lindoso Castle (Solé 2019) is worth highlighting, and in France, all the educational work around the new Guédelon Castle in Treigny, where, for over 20 years, they have been building ex novo strictly using materials, techniques, and procedures dating from the medieval period (twelfth and thirteenth centuries). The project was conceived as an open-air museum rooted in education and experimental archaeology, even though it is also an incentive to create jobs and a tourist attraction beyond compare; it is now an immense living museum. Several reenactments have been held in Italy for years inspired by both the Roman past and prehistoric cultures. Many projects promoted by the Emilia-Romagna region, and present on its website, such as Antiqva Italia, have tried to encourage a connection and training between involved operators, schools, and groups to implement good reenactment practices.

In Spain (Español-Solana 2019c), the best-known project of this kind is possibly "Open for works" by the Santa Maria Cathedral Foundation, in Vitoria (https://www.catedralvitoria.eus/). This is not a historical reconstruction model *per se*, but it successfully echoes the philosophy of this new dissemination and investigative trend in heritage. There are other important projects that schools are gradually recognizing as valuable means for history and heritage education, for example, Tarraco Viva Festival, which takes place in Tarragona every year, and Emerita Lvdica, in Mérida (Figure 2). The former was devised as an attraction for Tarragona's candidacy to become a World Heritage Site and it is now the city's most important heritage management project and, undoubtedly, the best known and most imitated in Spain. It attracts almost 200,000 visitors every year and its main purpose is to spread awareness of the Roman legacy on the peninsula. In Mérida, over 18 reenactment associations gathered in 2021, involving the local population and attracting

many visitors from elsewhere. These are just a few general examples that provide clues on how to tackle any educational project through organizational complexes, mostly managed by local authorities, in which historical reenactment is the primary educational technique.

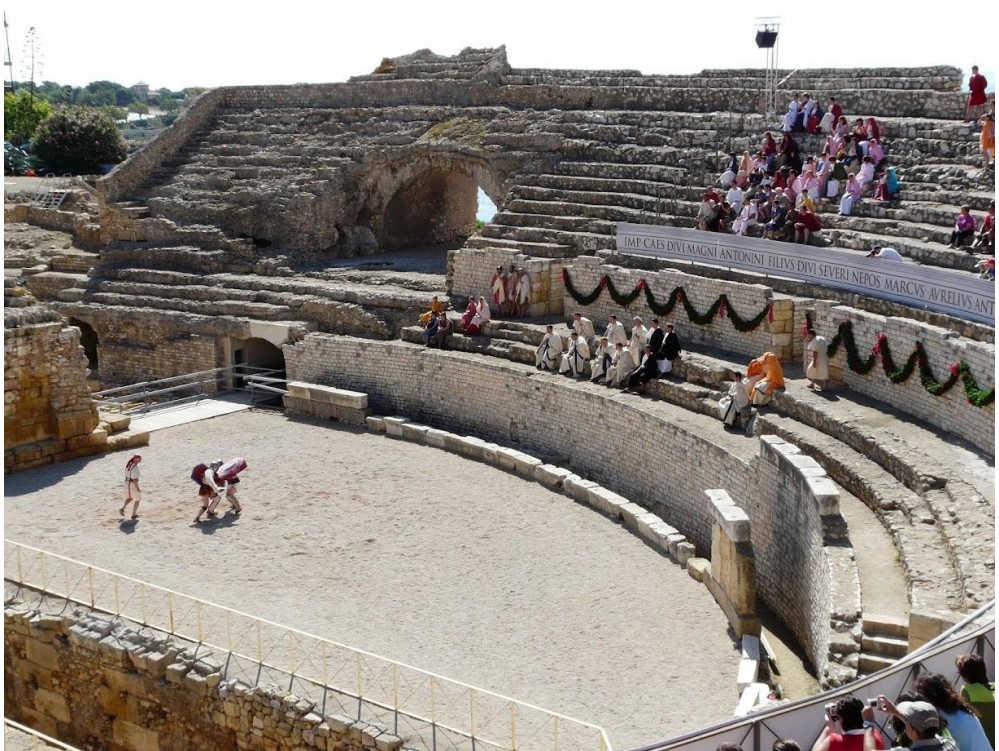

**Figure 2.** Tarraco reenactment, Spain.

Peracense Siglo XIII, also in Spain, exhibits the link between historical reenactment and cultural heritage (Figure 3). This interesting reenactment focuses on studying, researching, and recreating the clothing, weapons, and ways of life of the former territory of the Crown of Aragon in the thirteenth century. Other activities are also held in Peracense, which fall under the umbrella of live interpretation (Solé 2019) involving face-to-face contact between reenactors and visitors.

### 2.4. The Metaconcepts of History Instruction, the Key for Historical Reenactment

There are values in history that are sometimes hard for teachers to explain and even harder for pupils to understand (Sandwell and von Heyking 2014). It is usually said that history helps understand the present and avoid repeating past "mistakes", but, in truth, society's thinking about current aims, problems, and challenges is not well-formed. This is one of the reasons why the discipline is now distancing itself from the very society it is meant to serve. Consequently, history needs to be given value "so that students find something truly valuable there that transforms how they think about the world and themselves" (Paricio Royo 2018, p. 233). The benefits of history instruction for society are more than evident with the change in methodology that helps develop critical thinking at the service of democratic citizens. This is not something people acquire naturally as they mature psychologically (Stueber 2002); instead, they need to learn a series of skills that have to be worked on in teaching (Gómez Carrasco et al. 2014; Soria López 2015). There is also an ethical, behavioral, and attitudinal learning that is extremely useful in molding personality and for community social life. The difference between this type of history and history learned by accumulating and memorizing information is significant.

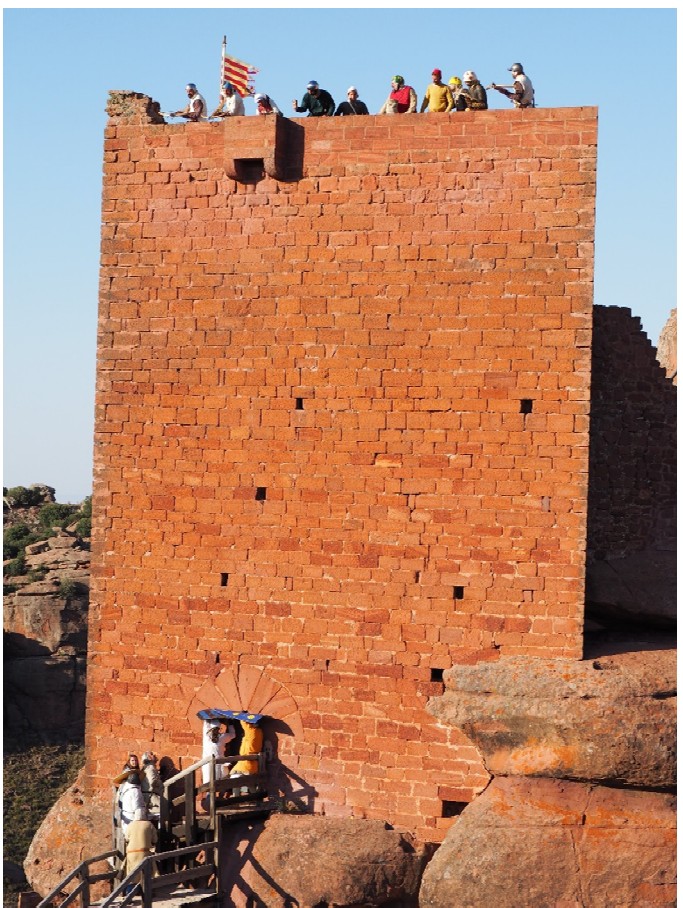

**Figure 3.** Peracense reenactment, Spain.

Historical thinking allows individuals to approach history independently and construct their own representation of the past as it gives them a series of instruments to analyze, understand, and interpret this history (Rüsen 2005; Santisteban Fernández 2010). It is a creative process limited to interpreting the sources historians produce to create reliable historical narratives. Some concepts are key to developing these narratives: sources, historical relevance, causation, change and continuity, historical perspective, and the ethical dimension of history (Lévesque 2009; Seixas and Morton 2013; Paricio Royo 2020). Soria López (2015) considers that the conceptual proposal put forward by Santisteban, González, and Pagès—although her statement is also applicable to Seixas and Morton's—is a dynamic model consisting of elements that are all crucial and configured in relation to the others, even though it is not a sequence of steps to follow. Citizens need a reflective and critical view of the past so they can be aware of their actions and take responsibility for them. This is far from memorizing dates and historical content, a practice followed by some teachers, which does not entail any reflection or interpretation.

That is why history learning should be focused on research, inquiry, and problems with individuals needing to deal with historical facts and develop cognitive skills and procedures for understanding and explaining them (Cardona Gómez and Feliu Torruella 2014; Paricio Royo 2018). "Unfortunately, schools are not usually designed to acquaint students with such investigative processes. Usually, curriculum structures and instructional patterns are designed to transmit content (which students are expected to remember) or to teach isolated skills (which students are expected to practice)" (Barton 2017, p. 459).

Teachers/historians, let alone their pupils, no longer limit themselves to talking or writing (Plá 2005), even though this has mainly been the way of capturing thinking so far. Today, it is usual for teachers and historians to become involved in multiple practical and multimedia actions enabling pupils or the interested general public to know

more and understand better by reflective thinking (Sandwell and von Heyking 2014; Franco-Calvo 2021).

Historical relevance involves reflecting on what and who should be remembered and why and, therefore, what and who should be studied. To this end, we need to analyze the social impact of the process, the importance contemporaries gave to it, how long the consequences lasted and continued, and what it reveals about our present (Seixas and Morton 2013; Rivero Gracia and Pelegrín Campo 2019). When a group decides to recreate a specific historical period, a choice has been made, determined by a variety of reasons. From then on, the group engages in an ongoing exercise of historical relevance: What do we want our reenactment to show? Which type of characters will appear in it? Which content do we want it to have and how? Should we focus on major events or everyday life? The reenactment generates a story of the past in which characters, events, and material culture all play a role in the historical narrative (Rüsen 2005).

"Given that historical reenactment is not a historical source, historical contents must be worked on using sources" (Jiménez Torregrosa and Rojo Ariza 2014, p. 38). The use of historical sources is significantly complex, since we cannot address them if we are incapable of interpreting their partiality or intentionality (Rüsen 2005). For that reason, we must be able to classify the sources based on their types, but also to analyze, evaluate, or compare them with other evidence (Barton 2017) so that we can use them to formulate hypotheses or solutions to the problems we have set ourselves. The rigor required in historical reenactments leads to using these sources to, as far as possible, document the event or activity to be recreated based on the information provided both in written documents and by archaeologists, who establish first-order evidence for reenactors in museums and cultural institutions. In general, there is a shortage of information that archaeology cannot fill. However, we must also consider that countless questions can be asked about these archaeological objects, and these questions are not exclusively concerned with the objects themselves, but also with their context, which includes the people that used them (Egea Vivancos and Arias Ferrer 2018).

Change and continuity are essential concepts shaping history in a constant back and forth, a coherent story about how the world transformed. The aim is not merely to point out changes between two points in history, but to link a narrative that establishes connections so that these changes make sense.

> "An all-encompassing, yet in-depth view of not only change processes but also of diverse and sometimes opposing forces that have led to these changes, and of resistance and continuity factors, as well as the meaning of these series of changes and the very nature of change in history". (Paricio Royo 2018, p. 236)

As Paricio Royo (2018) also points out, this involves selecting the most relevant processes, but from a dynamic stance in which changes condition future actions, thus fostering new historical transformation processes with a display of forces that try to promote or slow down said processes. But we must also explore how this change manifests itself and how it is perceived by contemporaries, while analyzing everything remaining from the past and shaping the present, straddling both realities. We cannot ignore that all this contributes to gaining historical perspective—historians' distance from events—when attempting to explain the whys and wherefores of the past, the causes and consequences, and how events were viewed by their contemporaries (Barton 2017; Paricio Royo 2020).

A historical reenactment is an effective tool for appreciating changes in everyday objects that, by using similarities and differences, allow us to place their owners in time and space (Egea Vivancos and Arias Ferrer 2018). We need historical consciousness as it will help us assess and interpret these changes and continuities while promoting understanding of concepts such as periodization or simultaneity (Rüsen 2005). Historical consciousness is the consciousness of time as it is based on the relationships we establish between the present and the past (Santisteban Fernández 2010). Acquiring historical consciousness means accepting our situation in the world at a specific moment in the development of

history we find ourselves in, where, based on that dynamic conception, we accept how fleeting our existence and way of life are (Paricio Royo 2018; Retz 2018).

The main instrument used in memorizing history is the unidirectional narrative. To escape this encyclopedic view, we have to break away from that use of narrative and take advantage of it to make the past understandable, while tackling problems linked to thinking historically (Rüsen 2005; Soria López 2015). Designing a historical narrative, in this case for historical reenactment, is no easy task if our aim is to develop historical thinking. Constructing a historical narrative involves using sources to produce a story that must be structured on an explanation of causes and consequences, which naturally involves seeking those causal connections. Why do events happen and what are their consequences?

The concept of historical empathy refers to the capacity to understand the attitudes and motives of past figures (González Monfort et al. 2009; Endacott and Brooks 2018; Paricio Royo 2019). Empathic understanding allows us to imbue other ways of life, experiences, norms, or past belief systems with meaning by putting ourselves in someone else's shoes and thereby explaining their decisions. There is a debate on whether emotional involvement is necessary or not in historical empathy exercises (Doñate Campos and Sarria 2019): defenders of improvement in the capacity to understand the emotional component against those remarking that this emotional involvement is not a form of historical reasoning (Paricio Royo 2019). Consequently, some confusion arises between the ambiguity of the terms empathy and sympathy. When conducting a rational study of the past, with no affective identification with the protagonists, or even understanding of events from an emotional standpoint, establishing the concept of historical perspective-taking was deemed necessary (Barton 2017; Paricio Royo 2020). This involves recognizing our own attitudes, beliefs, and intentions in a different historical and cultural context. We need to be able to differentiate between the past and the present by establishing a temporal distance that helps us recognize the reasons behind the actions (González Monfort et al. 2009), thereby overcoming presentism. Therefore, although empathy allows us to work on historical imagination, for it to have historical value, it must go hand in hand with contextualization (Santisteban Fernández 2010), with sources and evidence as its only starting point (Barton 2017; Paricio Royo 2019), and without forgetting that it is a process, as described by Foster, Ashby, Lee, and, more recently, others (Endacott and Brooks 2018, pp. 206, 209).

Based on these arguments we can ask ourselves: Is this not the aim of a historical reenactment? Of course it is. History can be learned by developing historical empathy through role-playing and hot-seating activities (Paricio Royo 2019), simulation (Hernàndez Cardona 2001; Kneebone and Woods 2014), recreating processes, or designing a reenactment (Rivero Gracia and Campo 2015), connecting with the more human part of history (Fontal 2008; Cardona Gómez and Feliu Torruella 2014). The more precise the historical reenactment is—approaching conditions similar to those in the past—the more empathic the experience will be. Can we really put ourselves in another's shoes? Can we feel what the protagonists of the events felt? This is one of the major problems, since empathy does not distinguish between cognitive and emotional elements. The answer to the above questions would obviously be "no" as this technique does not enable us to return to the past, but rather to interpret it (Rüsen 2005; Solé 2019). Although we are momentarily replicating an event, the society we live in is different from the one at that time, which is why our responses to stimuli will be too. Feeling safe despite the possible risks our actions may entail, and the possible presence of an audience, prevent us from thinking and acting as they did in the past, which distorts our reactions. We are limited by our way of understanding the world and we relate with others based on our own system of beliefs (Jiménez Torregrosa and Rojo Ariza 2014). But, without a doubt, historical reenactment is the best way of trying to experience history as it clarifies many of the questions we may have (Español-Solana and Franco-Calvo 2021b), always from a rational standpoint, although without excluding emotions and experiences (Paricio Royo 2019).

The ethical dimension, or historical consciousness, involves the present having its origin and acquiring significance in past eras, with societies in constant flux and individuals

playing a role, so the past is part of that individual (Gómez Carrasco et al. 2014; Retz 2018). Can history help us experience the present? Historical reenactment can perform an extremely important function in historical consciousness by highlighting the changes that occurred in the past, showing how contemporaries might have experienced them, and the consequences of their action or inaction, so that we can accept the historicity of the world and our own reality in the present and future possibilities.

Some historical reenactments not only focus on the ways of life of the elite or events such as battles that are represented, everyday life is also shown, by approaching mentalities, culture, and economic activities (Felices de la Fuente and Hernández Salmerón 2019).

If the basic elements of historical thinking consist of considering historical problems based on their relevance and impact on societies, with historical consciousness and historical reasoning founded on a critical use of sources (Retz 2018; Gómez Carrasco et al. 2014), duly contextualized historical reenactment can undoubtedly help develop the typical skills of historical thinking (Felices de la Fuente and Hernández Salmerón 2019). A good reenactor will implement what we can call reenactment thinking, which involves a "reenactment method" (Cózar Llistó 2013; Español-Solana and Franco-Calvo 2021b) and a way of working that leads us to develop historical thinking. Taking the steps that Jiménez Torregrosa and Rojo Ariza (2014) establish for historical reenactment as a reference, we can observe the important relationship between the key concepts established to achieve historical thinking. Consequently, historical relevance determines the contextualization steps for the episode, including the selection of the historical moment and the event to be recreated, and the choice of characters to be protagonists and secondary figures. Concepts such as change, continuity, and causation are present in the timing stages and in the choice of setting and propos within the narrative to establish, in which historical time will also play an essential role. Historical perspective and historical consciousness are connected in order to answer questions that help understand historical empathy such as: How did the protagonists feel? How would we feel and behave in similar situations? How do the decisions made impact the present?

## 3. Final Reflections and Conclusions

As mentioned above, the concept of historical thinking has been one of the newest and most interesting approaches in history instruction since the end of the twentieth century. Its origins are in the English-speaking world, in works by authors such as Dickinson, Lee, and Denis Shemilt. Later, it continued in the United States (Wineburg and VanSledright) and in Canada (Seixas and Lévesque), with slight variations between them. Interesting theoretical approaches are also being produced in Spain by Domínguez, Santisteban, Gómez, and Paricio, who have helped its use become more widespread.

The possibilities of historical reenactment in formal and non-formal education are immense. Its use involves applying methods that we could call new but have been around in practice for years. Historical reenactment, either observed or designed by pupils, allows them to develop a type of thinking that we have termed reenactment thinking, which, in turn, helps them develop historical and critical thinking.

This way of working on and analyzing history as a professional team leads to significant, contextualized, active, and dynamic learning, set in heritage sites, and it creates democratic citizens who opt to appreciate history and past culture for the values they contribute to society.

Consequently, as they are experiential, historical reenactments are becoming an effective didactic tool at historical monuments, and they are fostering increased interest at universities due to their accurate portrayals, alongside other characteristics such as the motivation, emotion, and the quality they achieve or their contribution to disseminating heritage. The tourism that takes place in these heritage spaces can have a huge economic, sociocultural and environmental impact, so it is necessary to establish a sustainable tourism that is compatible with the precepts established by the 2030 Agenda. We believe that historical reenactment can, perfectly, meet those criteria established with the 17 SDGs (specifically

04, Quality Education; 05, Gender Equality; 08, Decent Work and Economic Growth; 11, Sustainable Cities and Communities; & 12 Responsible Consumption and Production) and become a good practice for our communities and students.

In short, a good historical reenactor will be able to evaluate the relevance of events and the characters involved in them; manage to put themselves into past figures' shoes to understand their motives and actions; analyze the causes and consequences of the events; gain a perspective of the facts by considering alternatives to what happened—all in meticulous detail through a historical narrative created using sources and evidence and based on the research of material and intellectual culture. From these reenacted visions, the public will learn history more completely, more effectively, and more lastingly. There is no denying the contribution reenactments make to our discipline: teaching and learning History.

**Author Contributions:** All authors have contributed equally to the writing of the article. All authors have read and agreed to the published version of the manuscript.

**Funding:** Investigation Group S50_20R: ARGOS, Aragón Government, Spain. Grupo de Investigación S50_20R: ARGOS, Gobierno de Aragón, España.

**Institutional Review Board Statement:** Not applicable.

**Informed Consent Statement:** Not applicable.

**Acknowledgments:** Article written as part of the work performed by Argos Group, from the IUCA Institute of Research, University of Zaragoza, Spain.

**Conflicts of Interest:** The authors declare no conflict of interest.

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
