# Peer review of "Educating in History: Thinking Historically through Historical Reenactment"

_socsci, doi:10.3390/socsci11060256_

Round 1

Reviewer 1 Report

Some of my remarks have been indicated in the file attached. In my opinion the text must be improved before publication. The main concern is about the quality and clarity of the structure of the text. It is not logically connected in many parts. Moreover, there are a lot of ideas that are not connected to each other. This is the reason why sometimes the reader is lost in argumentation.

Also, the perspectives of a teacher, historian, reenactor, pupil or visitor are loosely connected. For instance, in one sentence it is said about the teacher, in another about reenactor and so on.

The same situation is with concepts like history, the past, heritage that are different in meanings, but used in the article as changeable concepts. The same is with reenactments, reconstructions and heritage sites. 

The choice of examples is quite unclear for me (v. 353 - 436). It is a simple  description and enumeration of examples from different periods, continents and of different forms. It makes an impression that it is a very accidental list with no reference in the text. 

Also, the methodology applied in the article should be clearly defined and described. 

640 – the authors refer to Agenda 2030 – I can ask why it appears in the conlusions when it was not even mentioned in the body text?

To sum, in my opinion it needs to be thoroughly rearranged and logically structured since in the version reviewed it is difficult to follow the main objective of the article. 

Author Response

Thank you very much for your considerations and improving ideas for the article. We have seen that you have read it with interest and we value hugely the time and effort you did. We are going to finish the revision today and hope we have collect all the suggestions. 

Reviewer 2 Report

This is a very interesting article arguing for the use of historical reenactment in the classroom and as a contribution in itself to historical consciousness. The points below would make the article clearer. 

Line 4 - is 'variables' the word you want here.

Lin 12 - 'in the area' should be changed to 'an area' or 'local area' or 'region'.  you are not talking about one area here. 

Line 21 leave out first four words  - begin with the most.

I think on line 33 you should clearly outline your several objectives and how you answer them. 

Line 65 As has been mentioned, leave out elsewhere.

115 - 124 take out of dot points, write in sentences.

139 facts

169 -184 do you have example of classroom use can you put it in or perhaps suggest one.

214-217 leave out you have said this

244 - you need concluding paragraph for this section.

318 - clearly distinguish between classroom and public performance here , classroom and heritage site need to be clearer 

333 do you mean quality mark - this sentence needs to be rethought. 

468 - memorizing content is not modern curriculum but you might say teachers fall back into such a mode of teaching.

481 -3 leave this sentence out.

Author Response

Thank you very much for your considerations and improving ideas for the article. We are going to finish the revision today. 

Reviewer 3 Report

The review presented shows the importance of a topic that is increasingly more prevalent and with a growing interest for the academic research.

This review is clear and of relevance to the field and it includes relevant current citations.

It is consistent between all its parts and shows a clear common thread to justify the importance of historical reenactment in formal and non-formal education to build the historical thinking.

The conflicts around the historical re-enactment also show that is really important to differenciate between historical festivals and historical reenactments, in the strict sense, in the Spanish context. But the most important aspect of this review is to show the posibilities of the historical reenancement. As the authors show in the paper, it is really important to question ourselves about how we think historical reenancement is.  

Conclusions drawn are coherent and supported by the mentioned citations.

Author Response

Thank you very much for your considerations. 

Round 2

Reviewer 1 Report

Some changes have been made by the authors and now the paper is more clear I think. 

This manuscript is a resubmission of an earlier submission. The following is a list of the peer review reports and author responses from that submission.

Round 1

Reviewer 1 Report

Dear Authors ,

You have prepared an extensive and interesting text, but I have a few reflections about the structure of the article and the methodological layer.

  1. Please adapt the structure of the article to the requirements of the MDPI. “The structure should include an Abstract, Keywords, Introduction, Materials and Methods, Results, Discussion, and Conclusions (optional) sections, with a suggested minimum word count of 4000 words”. More information here : https://www.mdpi.com/authors/layout
  2. First of all, there is no methodological chapter (Materials and Methods ) in which you will describe the research procedure used and the purpose of the research. Typically, the PRISMA model is used in the literature review. You can find some information here ( Systematic Reviev ): https://www.mdpi.com/about/article_types
  3. Results chapter is also missing (everything was included in the Discussion chapter).
  4. Mistakenly formulated aim of the research : " This paper aims to review the scientific literature relating historical thinking to historical 8 reenactment to try to verify the link between both variables." [ line 8]. The aim can never be a "literature review" (just as the goal of research cannot be to conduct interviews or a survey). In the case of your article, the goal may be, for example, to identify trends, develop a definition catalog, etc.
  5. The conclusion should contain a direct reference to the purpose (s) of the research.
  6. Among the keywords there is "sustainable tourism" meanwhile, this thread does not appear in the text at all (except for line 653).
  7. Please consider the appropriateness of the chapter “2.3 Some examples of historical reenactment worldwide ”- the topics discussed here are loosely related to the topic of the article, and the examples themselves also seem to be quite random.

Good luck.

Author Response

Thank you for your comments

Reviewer 2 Report

Interesting paper on historical reenactment and the usefulness of thinking historically throughout the whole process.

The flow of the overall argument could be improved. An overall revision of the structure of the argument, and clarification of the concepts used would be interesting.

I would suggest first to better define what the authors mean by thinking historically. Second, there seems to me to lack a discussion on critical thinking throughout, given that most reenactment focuses on what is more visible. 

The notion of empathy could be deepened here with respect to critical analysis.

Some concepts need more definition such as “second order historical concepts”, especially for a journal that has not so much attention to historical concepts. On page 3, I have commented that you “should explain what are these specific cognitive skills in your discipline. It would not be obvious for the reader of this journal not familiar with historiography”.

Section 2 should be improved in terms of structure or the argumentation. 

The notion of reenactment thinking should be further developed.

The conclusion could be more developed, retaking the arguments - or some of them - in the previous section 2. 

Some authors are difficult to find in the references such as d’Oro 2004. De certau is not written correctly - see de Certeau 1984. 

The appearance of the bibliography seems to follow the order of appearance in the text, but it is not always the case. Why do you use numbers in the list of references but not in the text. Reading the paper it is hard to figure out if your list of references is complete and it is hard to find the reference that is cited in the texto. You should improve this. Or alphabetical order of the reference list or a number associated to each reference in the text that can be found in the Reference list section.  

Author Response

Thank you for your comments

Reviewer 3 Report

The major weakness in what is otherwise an interesting paper is that it lacks relevance to the Journal ‘Sustainability’. Scant references to ‘sustainable tourism’ and a couple of sentences in the conclusion do not make up for the fact that this is a history paper or a teaching paper but really has no evidence of the relevance to the journal’s theme. However, if the authors could review the paper’s content and make it more relevant, it would be interesting and useful.

Specific feedback follows:

Abstract

How does this paper relate to sustainability - this should be made clear with a rationale that is prominent within the Abstract, not just a mention in Line 19

Avoid informal idioms / non-academic language e.g. Line 13 ‘is a must’

Is the literature ‘scientific’ or ‘academic’?

Line 11 ‘needs’ should be ‘need’ - typo

Line 19 edit ‘and sustainable tourism’

The Abstract could identify the core purpose and rationale of the paper more explicitly and clear - it lacks logical flow

The paper looks fascinating

Introduction

Line 26 again, why ‘scientific’ - is ‘academic’ more appropriate?

I would hope to see a definition of the phrase ‘living history’ rather than the assumption that it is the same as reenactment. Then an exploration of why they are similar.

Line 37 is ‘academic support’ different from ‘scientific’?

Introduction

The Introduction should begin with a more explicit indication of the scope and aim of the paper, rather than going straight into a lit review of the characteristics of reenactment

Line 7 I think you need top define what you mean by ‘festivals’ - and their context

Lines 75-78 are you saying that ‘only’ academics can develop and deliver ‘authentic’ / ‘real’ reenactment?

line 98 clarify what you mean by ‘the latter’ - what does it actually refer to? 

The Introduction lacks any reference to ‘Sustainability’ / ‘Sustainable tourism’. The journal is ‘Sustainability’, therefore any paper should explicitly link itself to the theme throughout. Otherwise, what is the authors’ rationale for publishing in this journal as opposed to any other?

Lines 120-129 as well as listing the authors at least add a brief summary of their conclusions that can later be subject to critical discussion. It is not clear what the bracketed information adds - is it book titles?. Explain why each is relevant and what they add to the discussion - don’t assume that the reader knows (I don’t).

Line 134 what do you mean by professionals?

line 144 typo - facs should be facts?

T The he paper at this point seems to be more about teaching history - where is sustainability in the discussion?

Line 223 discuss the differences between professional and amateur reenactment - who does these? Is one better than the other? More accessible?

Line 227 surely reenactment also involves interpretation? What are the power structures involved in this?

Line 250 each section should have a brief summary

Line 253 typo - delete comma after hyphen

Line 338 Tourism is mentioned! Line 345 Sustainability is mentioned!

The authors could explore the importance of sustainability - in tourism; - in reenactment

Lines 371-372 The English Civil War Society would argue that The Sealed Knot lacks authenticity in materials and technique. Broth organisations do MUCH more than ‘dress in 17th century fashion’

Line 382 typo ‘took place’ not tools place’… the rest of that sentence does not make sense in English. Did 2016 see the 20th reenactment?

The examples are purely descriptive, this section lacks evidence of critical analysis, empirical research and even much detail. It could be a really interesting section. 

Refer to and explain relevance of photographs in the text

The quote in lines 481-485 should be in a new line, indented and italicised like other long quotes

Line 577 Foster Ashby Lee (year/s?)

Conclusion

Lines 633-636 should include (year/s) by authors

line 652-653 Sustainable tourism is mentioned - this section which is really relevant mentions the 2030 Agenda (explain what this is, add reference); 17 SDGs (explain what these are, add reference) - neither are mentioned previously in the paper 

The paper lacks a critical discussion of sustainability elements

Needs integration of sustainability theme, not sparse mentions

Line 663 what is your discipline - this is relevant in an interdisciplinary context

Author Response

Thank you for your comments